# Inaugural Description of Extrafloral Nectaries in Sapindaceae: Structure, Diversity and Nectar Composition

**DOI:** 10.3390/plants12193411

**Published:** 2023-09-28

**Authors:** Danielle Maximo, Marcelo J. P. Ferreira, Diego Demarco

**Affiliations:** Departamento de Botânica, Instituto de Biociências, Universidade de São Paulo, São Paulo 05508-090, SP, Brazil; daniellemaximo@usp.br (D.M.); marcelopena@ib.usp.br (M.J.P.F.)

**Keywords:** nectar, ocelli, sugars, amino acids, ants, defence, *Urvillea*

## Abstract

Sapindales is a large order with a great diversity of nectaries; however, to date, there is no information about extrafloral nectaries (EFN) in Sapindaceae, except recent topological and morphological data, which indicate an unexpected structural novelty for the family. Therefore, the goal of this study was to describe the EFN in Sapindaceae for the first time and to investigate its structure and nectar composition. Shoots and young leaves of *Urvillea ulmacea* were fixed for structural analyses of the nectaries using light and scanning electron microscopy. For nectar composition investigation, GC-MS and HPLC were used, in addition to histochemical tests. Nectaries of *Urvillea* are circular and sunken, corresponding to ocelli. They are composed of a multiple-secretory epidermis located on a layer of transfer cells, vascularized by phloem and xylem. Nectar is composed of sucrose, fructose, xylitol and glucose, in addition to amino acids, lipids and phenolic compounds. Many ants were observed gathering nectar from young leaves. These EFNs have an unprecedented structure in the family and also differ from the floral nectaries of Sapindaceae, which are composed of secretory parenchyma and release nectar through stomata. The ants observed seem to protect the plant against herbivores, and in this way, the nectar increases the defence of vegetative organs synergistically with latex.

## 1. Introduction

A nectary is a gland that secretes nectar, a secretion mainly composed of sugars [1]. The fact that the concept of nectary is functional implies a great morphological diversity. Nectaries can be trichomes, epidermal surfaces, as well as glands composed of nectariferous and subnectariferous tissues, which can be vascularized or not [1,2].

The type of nectary found in a given clade is directly related to the morphological evolution of secretory structures of this group and their ecological relationships. This fact can be especially noted in Sapindales, a morphologically multidiverse order that shows an unusual diversification of their glands [3].

Different types of nectaries have already been reported for Sapindales, an uncommon diversity for the same order. Moreover, extrafloral nectaries have been recorded only in Neotropical lineages, i.e., Anacardiaceae, Burseraceae, Sapindaceae, Simaroubaceae, Meliaceae and Rutaceae [3,4,5,6,7,8,9,10,11,12], and data on their anatomical structures are rare, creating a huge gap of knowledge, especially for nectaries of Sapindaceae.

Despite the large number of species in this family and the morphological studies of its representatives, the first report of extrafloral nectaries in Sapindaceae has only occurred recently in *Paullinia*, *Serjania* and *Urvillea* [3], and there is no structural information about them. Extrafloral nectaries vary in shape and size in the different families of Sapindales and could be trichomes, stalked glands or ocelli located on leaves, cataphylls, bracts, inflorescence axis and fruits [3,4,5,6,7,8,9,10,11,12,13].

Despite the morphological and ontogenetic variation, all nectaries recorded in Sapindales are composed of nectariferous parenchyma, vascularized by xylem and phloem, and release nectar through stomata occurring in a non-secretory epidermis [3,12,13]. Although nectaries of Sapindaceae would supposedly have a similar tissue composition [3], there is no anatomical information to date.

Previous personal field observations indicate a relationship between the nectar produced by these nectaries and the attraction of a large number of ants that patrol the young branches, as reported by Villatoro-Moreno et al. [14] in a largely well-known mutualistic relationship [15,16,17,18,19]. Although nectar is composed basically of water and sugar, other minor components may occur in this secretion, such as amino acids and proteins, lipids, and even phenolic compounds and alkaloids [1,20,21,22]. There is no information about nectar composition in Sapindaceae. The analysis of secretion composition is fundamental to correctly classify a gland and perform ecological inferences [23].

Due to the scarcity of information about the extrafloral nectaries in Sapindaceae and their likely distinct constitution from all other nectaries of Sapindales, the goal of this study was to analyse the structure of extrafloral nectaries and the main chemical classes of metabolites in *Urvillea* nectar to better understand their morphological, anatomical and functional diversity in Sapindaceae.

## 2. Results

### 2.1. Morphology

The leaves of *Urvillea ulmacea* are trifoliolate with leaflets exhibiting a dentate margin (Figure 1A) and nectaries in the tooth apex (Figure 1B). These nectaries are inconspicuous, forming a small submarginal rounded bulge on the abaxial surface of the leaflets (Figure 2A), where the nectariferous tissue is found slightly sunken, corresponding to an ocellus (Figure 2B). This secretory epidermis is formed of cells with an irregular pattern of organization, covered by a smooth cuticle, and there are no stomata (Figure 2C,D).

### 2.2. Anatomy

Anatomically, *Urvillea* nectaries are comprised of a multiseriate secretory epidermis (Figure 3A) covered by a thin cuticle, with three to six layers of cells located on a layer of transfer cells (Figure 3B). The secretory cells are isodiametric and have thin, pecto-cellulosic walls, dense cytoplasm with several vacuoles full of secretion (Figure 3C), and a prominent nucleus with evident nucleolus. These cells are juxtaposed and irregularly grouped since the beginning of their ontogeny (Figure 3D). However, these cells split, forming deep slits in the secretory tissue during the secretory phase (Figure 3B,E), indicating a likely relation to the release of nectar, which reaches the nectary surface without rupture of the cuticle (Figure 2D or Figure 3C). The transfer cells, beneath the secretory epidermis, have thick, primary (Figure 3C,F) and suberized anticlinal walls (Figure 4A,B) and a single central vacuole containing phenolic compounds (Figure 3C).

The nectaries are vascularized by ramifications of the vascular bundles of the leaf tooth (Figure 3A,B), composed mainly of phloem (Figure 3E) but also containing xylem vessels (Figure 3F). The vascular terminations expand beneath the transfer layer throughout the nectary extension. The leaf mesophyll underlying the nectaries has a large amount of laticifers, phenolic (Figure 3A,B) and crystalliferous idioblasts (Figure 3F) containing druses in a similar distribution to the rest of the leaf.

### 2.3. Nectar

The histochemical analysis detected the presence of carbohydrates (sugars), proteins and/or amino acids and phenolic compounds in the nectar (Figure 4C–F and Table 1). No starch storage was found in the nectariferous or subnectariferous tissue. Lipids are present in the nectar but were not histochemically detected (Table 1) due to their very low concentration (Figure 5 and Table 2).

The chemical analysis revealed the nectar is mainly composed of sucrose, fructose, xylitol and glucose, in addition to some minor components, such as amino acids, lipids (Figure 5 and Table 2) and phenolic compounds (Figure 6 and Figure 7). Comparing the phenolic derivatives of nectaries and leaves, four flavones were found exclusively in the nectaries of *Urvillea ulmacea* (Figure 6 and Figure 7; compounds 1–3 and 6).

After the release of nectar on the surface of the nectary, it is gathered by a large number of ants that forage the plant. Although nectar droplets have been rarely observed, the ants collect the secretion without damaging the nectary. During field collection of the plants, several leaves were analysed, and no wounds were observed.

## 3. Discussion

In this study, we unravelled the structure of extrafloral nectaries in Sapindaceae, revealing an unexpected gland formed by multiple epidermis, subtended by a layer of transfer cells, which has no parallel in Sapindales. Despite the large number of ants foraging leaves and shoots of several species of Sapindaceae (pers. obs.), the first reports of leaf nectaries in the family are recent [3,14], probably due to its greatly reduced size.

Extrafloral nectaries vary in position within Sapindaceae. They have been noted on the teeth of leaflet margin in *Paullinia seminuda* Radlk. [3], as observed in this study for *Urvillea ulmacea*. However, the nectaries were found at the leaflet apex in *Paullinia carpopoda* Cambess. [3] and in the apical region or along the midrib in *Nephelium lappaceum* L. [14]. The morphology of these nectaries also varies in the family. They are sunken ocelli in *Urvillea*, elevated ocelli in *Nephelium* [14] and stalked glands in *Paullinia* [3].

The structure of the extrafloral nectaries is highly diverse in Sapindales, but the histological composition described for those of *Urvillea* represents a novelty for the order. In Anacardiaceae, they are trichomatous, found clustered in depressions on the leaf surface and often confused with domatia [5,8]. In Meliaceae and Rutaceae, nectaries are ocelli formed by nectariferous epidermis and parenchyma [4,6,7,9,10,11]. Although Simaroubaceae also has ocelli on the surface of the leaf blade, nectaries formed only by secretory parenchyma, which release nectar through stomata, were recently observed at the apex of *Homalolepis* and *Simaba* leaflets [3,12].

Unlike the other families, Burseraceae and some Simaroubaceae have stalked nectaries, generally occurring on the leaves (petiole, blade and/or stipule) and cataphylls [13,24,25,26,27,28,29,30]. They are elongated, vascularized emergences, usually located on the margin of leaflets [12,31] but vary in *Ailanthus*, where they are found on the petiole (or stipule), lamina and margin of the cataphylls [13].

Although the nectaries of *Urvillea* stand out for being formed of a multiple epidermis with irregular cell arrangement, the irregularity in the organization of epidermal secretory cells is common in Sapindales. Glandular trichomes of Sapindaceae, Meliaceae and Simaroubaceae are characterized by the irregular arrangement of their secretory cells [3,12], which seems to be the pattern of cell organization in the members of these families, denoting an unusual activity of the protoderm during the ontogenesis of these glands. Furthermore, we observed that secretory cells, grouped in distinct clusters, split during secretory activity. These slits formed during nectary differentiation indicate that the nectar may be released laterally to the inner cells of the secretory tissue, similar to that observed in some colleters [32,33,34,35]. In *Plumeria* (Apocynaceae), the separation of the epidermal cells is due to the dissolution of the middle lamella [32]. More studies are needed to verify the mode of secretion release in nectaries in Sapindaceae and the formation of these slits.

The multiple secretory epidermis of *Urvillea* are situated on a layer of transfer cells. Despite the large number of cells that form these nectaries, the presence of a layer of transfer cells under the secretory portion is similar to that reported for nectariferous trichomes of *Adenocalymma* (Bignoniaceae) [36]. This transfer layer is associated with the short-distance transport of solutes via the transmembrane. Apparently, transfer cells occur when the area for transport is much smaller than the volume of the destination structure and the transported solute is accompanied by minimal solvent flux [37,38]. In *Adenocalymma* nectaries, cell wall invaginations are also located in anticlinal regions, and a large number of mitochondria are associated with the polarized cell transport of solute [36].

The absence of stomata in the nectaries of *Urvillea* is also an unusual feature for the order, except for the trichomatous ones [3]. The release of nectar directly through the wall and cuticle would not be expected even for the family since the floral nectaries of Sapindaceae are formed of secretory parenchyma that release the nectar through stomata [3,39,40,41,42,43,44,45,46,47,48,49].

The nectar of *Urvillea* is composed of sugars, amino acids, lipids and phenolic compounds. Sucrose, fructose and glucose are the main components of the nectar of most plants [1] and are among the major compounds of the nectar in *Urvillea*. However, a fourth sugar was found in large amounts—xylitol. The occurrence of this sugar in the nectar is unexpected since xylitol is not commonly produced by plants. Its presence may be related to the growth inhibition of bacteria provided by xylitol [50]. Further studies are necessary to investigate the production and functions of this sugar in nectar.

Phenolic compounds, commonly found in the nectar, also have antimicrobial activity [1,51]. The phenolics detected in this study were identified as flavones, which have these properties [52,53,54]. The presence of these compounds is very important in secretions composed basically of water and carbohydrates [55] to avoid proliferation of microorganisms that may cause necrosis of the nectariferous tissues. Additionally, flavonoids provide photoprotection to the secretion, preventing the oxidation of compounds. Therefore, the occurrence of flavonoids in the nectar of *Urvillea* may prolong the secretory activity of the leaf nectaries.

The second major class of compounds in nectar is amino acids, which account for about 14% in *Urvillea*. This high concentration of amino acids in the nectar is unusual. In general, amino acids are 100–1000 times less concentrated than sugars in nectar. However, a higher concentration of amino acids has a greater potential to attract ants since it influences the taste of nectar, increasing its attractiveness and its nutritional importance, especially for animals whose only food resource is nectar [56,57,58,59]. Perhaps the high concentration of amino acids in the nectar of *Urvillea* is one of the factors responsible for the large number of ants observed continuously foraging the plants. Although the formation of droplets on the surface of the nectary is rarely observed, it is likely stored in the slits of the epidermis, where the nectar is collected, especially in young leaves.

Extrafloral nectaries have a well-known mutualistic relationship with ants that protects the plant against herbivores [15,17,19,60], and this interaction in Sapindaceae has been studied by Villatoro-Moreno et al. [14], who reported 10 species from five subfamilies of ants visiting the nectaries of *Nephelium lappaceum*. The ants were observed continuously foraging *Urvillea ulmacea* in the field, effectively protecting the plant. This protection associated with the occurrence of laticifers and phenolic idioblasts [61] inhibits herbivory; no wounds were observed in the leaves.

## 4. Materials and Methods

### 4.1. Plant Material

Shoots of *Urvillea ulmacea* Kunth were collected from the Forest Reserve of the Cidade Universitária Armando de Salles Oliveira (CUASO) in São Paulo (Brazil) and the voucher was deposited in the Herbarium SPF (Maximo, D. 1). This species was selected based on previous observations that identified nectaries occurring on the margins of its leaflets [3].

### 4.2. Structural Analyses

Shoot apices and young leaves were fixed in formalin–acetic acid–alcohol (FAA) solution for 24 h [62] and buffered neutral formalin in 0.1 M sodium phosphate buffer (pH 7.0) for 48 h [63]. For the micromorphological study, mature nectaries were isolated, dehydrated in a graded ethanol series, dried by the critical point method, mounted on aluminium stub, and sputter-coated with gold, with subsequent observation in a Zeiss Sigma VP scanning electron microscope (Carl Zeiss, Oberkochen, Germany).

For anatomical analyses, shoot apices and young leaves were isolated, dehydrated through a tertiary butyl alcohol series [62], embedded in Paraplast^®^ (Leica Microsystems Inc., Heidelberg, Germany), and serial-sectioned at a 12 µm thickness on a Leica RM2145 rotary microtome. Longitudinal and transverse sections were stained with astra blue and safranin O [64] and the slides were mounted with resin Permount (Fisher Scientific, Pittsburgh, PA, USA). Observations and photographs were performed using a Leica DMLB light microscope.

### 4.3. Nectar Composition

#### 4.3.1. Histochemical Tests

The main chemical classes of the constituents of nectar were investigated using the following histochemical tests in the embedded nectaries: periodic acid–Schiff’s (PAS) reaction for carbohydrates [65], ruthenium red for acidic mucilage [66], tannic acid and ferric chloride for mucilage [67], Lugol’s reagent for starch [62], aniline blue black for proteins [68], Sudan black B and Sudan IV for lipids [69], Nile blue for acidic and neutral lipids [70], copper acetate and rubeanic acid for fatty acids [71,72], ferric chloride and fixation in ferrous sulphate–formalin for phenolic compounds [62], and Dragendorff’s [73] and Wagner’s [74] reagents for alkaloids. Standard control procedures were carried out according to Demarco [23]. The autofluorescence of the secretion and suberized walls were also analysed under UV and blue light. All observations and photographs were performed using a Leica DMLB light microscope equipped with an HBO 100 W mercury vapor lamp and a blue light filter block (excitation filter BP 420–490, dichromatic mirror RKP 510, suppression filter LP 515) and UV filter block (excitation filter BP340-380, dichromatic mirror RKP400, suppression filter LP425).

#### 4.3.2. Derivatization of the Sample and Identification of Compounds through GC-MS

The extrafloral nectaries of *Urvillea* were derivatized using methoxyamine hydrochloride dissolved in pyridine (28 μL, CAS 593-56-6, Sigma-Aldrich, St. Louis, MO, USA) for 2 h at 37 °C and N-Methyl-N-(trimethylsilyl) trifluoroacetamide (48 μL, MSTFA, CAS 24589-78-4, Sigma-Aldrich) for 30 min at 37 °C [75]. The metabolites were analysed by GC-MS equipped with the HP-5MS column (Agilent, length 30 m, ID 250 μm, 0.25 μm film thickness, Santa Clara, CA, USA). The initial column temperature was adjusted to 70 °C for 5 min and ramped at 5 °C min^−1^ to a final temperature of 320 °C, which was maintained for 8 min with a total run time of 58 min. The injection volume was 1 μL in splitless mode with helium as a carrier gas at 1 mL min^−1^. The injector, ion source, and quadrupole temperatures were 300 °C, 200 °C, and 280 °C, respectively. MS detection was performed with electron ionization (EI) at 70 eV, working in the full-scan acquisition mode ranging between 50–800 *m*/*z* at 2.66 scan s^−1^.

Compound identification was made by comparison of mass fragmentation using NIST digital library spectra (v2.0, 2008) employing comparison values of Match e R-Match above 900 as well as by retention time and mass fragmentation pattern of commercial standards.

#### 4.3.3. Crude Extracts Preparation and HPLC-DAD Analysis of Methanol Extracts

For phenolic extraction, fresh fragments of leaves and nectaries were extracted with 1 mL of HPLC-grade methanol for 15 min in an ultrasonic bath at room temperature followed by the collection of the supernatants by centrifugation (13000× *g*, 4 min, 25 °C). The obtained solutions were filtered using a 0.45 µm syringe filter.

Phenolic compounds were analysed using HPLC-DAD (model: 1260 system, Agilent Technologies, Santa Clara, CA, USA) equipped with an autosampler, using a Zorbax Eclipse Plus C18 column (150 × 4.6 mm, 3.5 µm particle diameter) at 45 °C with a flow rate of 1 mL·min^−1^, and an injection volume of 3 μL. The detection wavelengths were registered at 254, 280, and 352 nm. The chromatographic method was constituted by a gradient of mixtures of solvents A (water acidified with 0.1% acetic acid) and B (acetonitrile) of 0–6 min: 10%B; 6–7 min: 10–15%B; 7–22 min: 15%B; 22–23 min: 15–20%B; 23–33 min: 20%B; 33–34 min: 20–25%B; 34–44 min: 25%B; 44–54 min: 25–50%B; 54–60 min: 50–100%B.

## 5. Conclusions

In this study, we described the structure of extrafloral nectaries in Sapindaceae for the first time. *Urvillea ulmacea* have nectaries of the ocelli type composed of multiple secretory epidermises positioned on a layer of transfer cells, vascularized by phloem and xylem. This nectary stands out due to an irregular arrangement of the secretory cells that split, forming deep slits where the nectar is temporarily stored. Nectar is mainly composed of sucrose, fructose, xylitol and glucose, in addition to amino acids and other minor compounds, such as lipids and phenolics. After the release of nectar onto the gland surface, it is gathered by many ants, which continuously forage the plant. This novel description reveals a new type of nectary that differs structurally from all others of Sapindales and display a high concentration of amino acids, which can be related to a more effective attraction of ants. The redundancy in defensive secretory systems of *Urvillea* may be related to the low predation rate of its leaves, which contain extrafloral nectaries, laticifers and phenolic idioblasts. Further studies of nectaries of Sapindaceae are needed to verify whether the peculiar characteristics observed in this study are typical of the genus or family.

## Figures and Tables

**Figure 1 plants-12-03411-f001:**
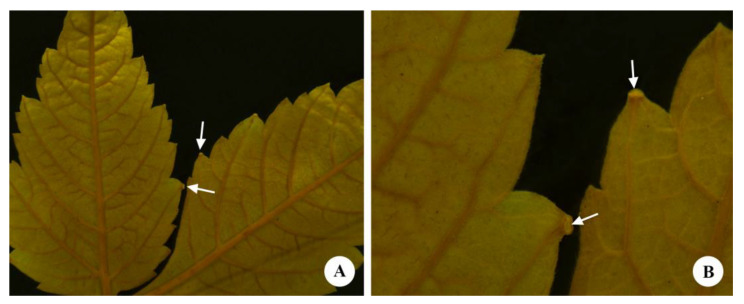
Leaf of *Urvillea ulmacea* Kunth. (**A**) Leaflets with dentate margin. Note nectaries in the tooth apex. (**B**) Detail of image A. (Arrow = extrafloral nectary).

**Figure 2 plants-12-03411-f002:**
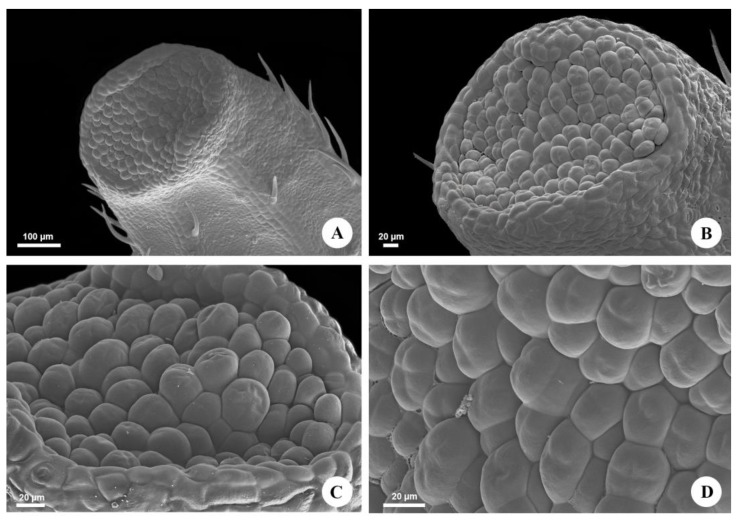
Micromorphology of the extrafloral nectaries of *Urvillea ulmacea* Kunth. Scanning electron microscopy. (**A**) EFN on the abaxial surface of the tooth apex. (**B**) General view of the nectary (ocellus). (**C**) Nectariferous epidermis composed of irregularly arranged cells. (**D**) Detail of the secretory epidermis. Note the smooth cuticle and the absence of stomata.

**Figure 3 plants-12-03411-f003:**
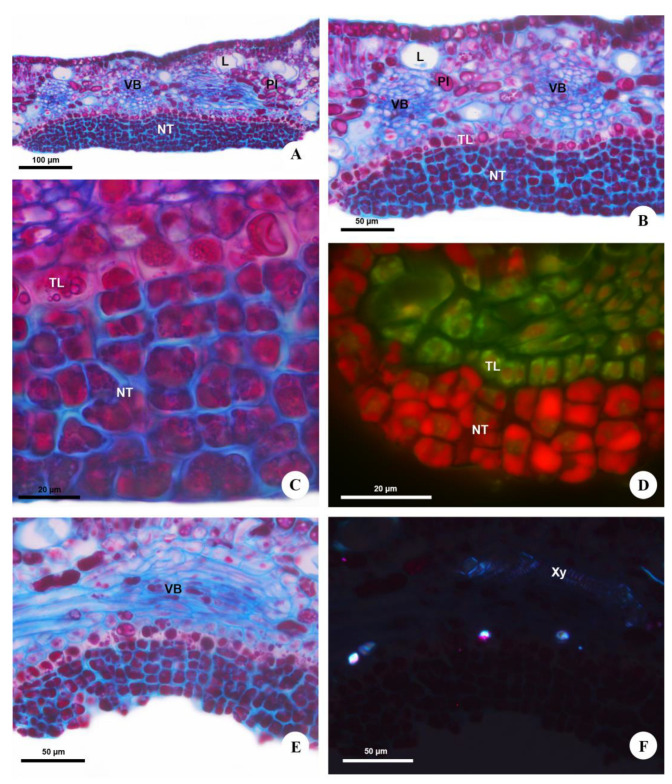
Structure of the extrafloral nectaries of *Urvillea ulmacea* Kunth. Light microscopy. (**A**–**C**,**E**) Bright field. (**A**) General view of the nectary on abaxial surface of the leaflet. (**B**) Nectariferous tissue situated on a layer of transfer cells. (**C**) Detail of the nectary. Note the secretory cells full of secretion, and the transfer cells containing phenolic compounds. (**D**) Developing nectary. Note numerous cells in different phases of cell division. Section stained with astra blue and safranin observed under blue light. (**E**) Vascular bundle of the nectary mainly composed of phloem. (**F**) Xylem vessels and crystals beneath the nectary evidenced by polarized light. (L = laticifer; NT = nectariferous tissue; PI = phenolic idioblast; TL = transfer layer; VB = vascular bundle; Xy = xylem).

**Figure 4 plants-12-03411-f004:**
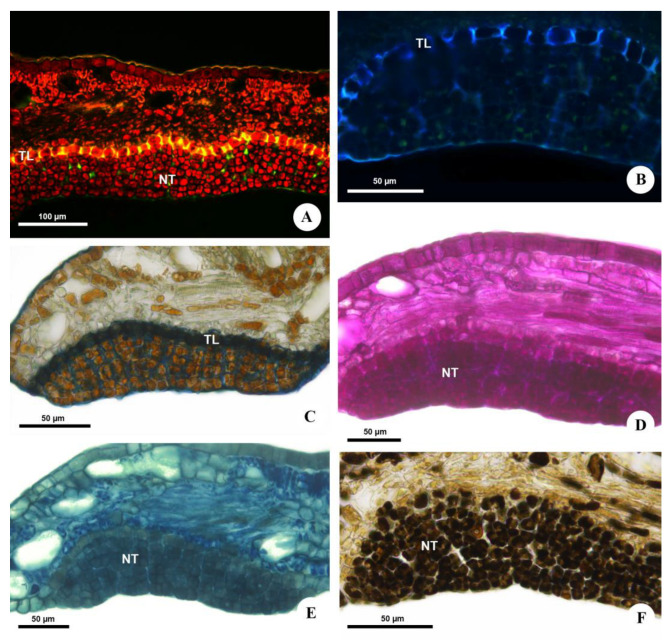
Histochemical analysis of the extrafloral nectaries of *Urvillea ulmacea* Kunth. (**A**,**B**) Fluorescence microscopy. (**C**–**F**) Bright field. (**A**,**B**) Suberized walls of the transfer cells evidenced by yellow fluorescence under blue light (**A**) and blue fluorescence under UV (**B**). Sections stained with astra blue and safranin. (**C**) Suberin of the transfer cells stained with Sudan black B. (**D**) Carbohydrates detected by PAS reaction. (**E**) Proteins and/or amino acids identified using aniline blue black. (**F**) Phenolic compounds evidenced by ferric chloride (NT = nectariferous tissue; TL = transfer layer).

**Figure 5 plants-12-03411-f005:**
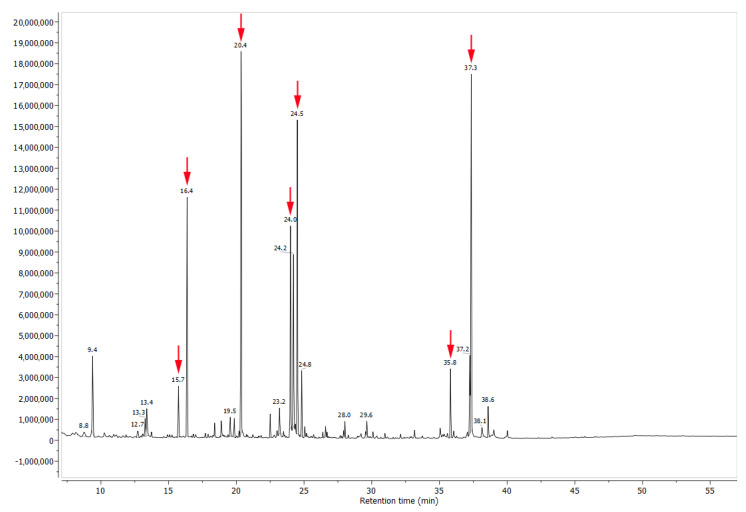
Chromatogram of compounds detected in the nectary of *Urvillea ulmacea* Kunth. Compounds exclusively detected in the nectar (arrows) are described in Table 2.

**Figure 6 plants-12-03411-f006:**
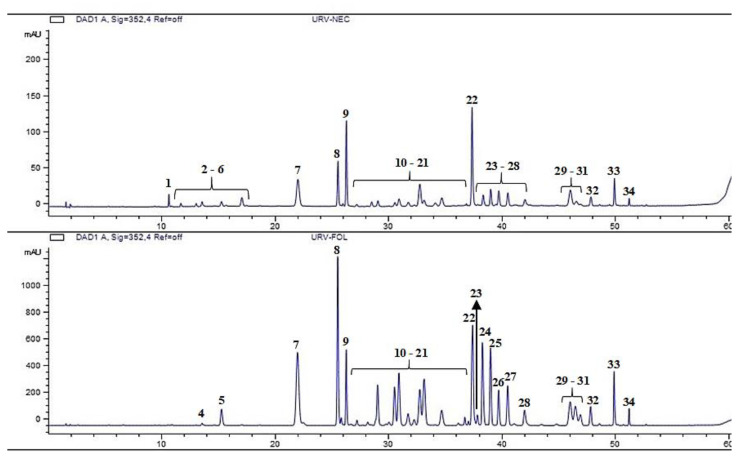
Chromatograms detected at 352 nm from the methanol extracts of nectaries (**above**) and leaves (**below**) of *Urvillea ulmacea* Kunth.

**Figure 7 plants-12-03411-f007:**
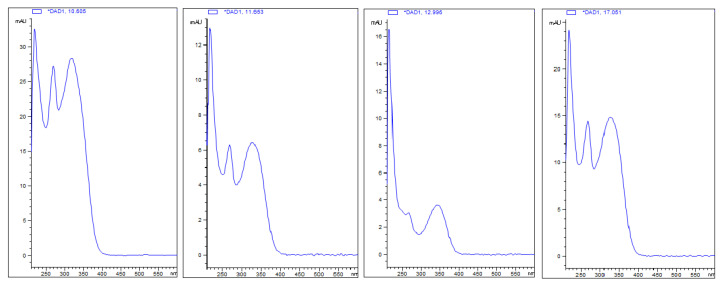
UV spectra of phenolic compounds (1–3 and 6) found exclusively in the nectaries of *Urvillea ulmacea* Kunth.

**Table 1 plants-12-03411-t001:** Histochemical tests applied to extrafloral nectaries of *Urvillea ulmacea* Kunth to identify the chemical classes of metabolites that compose the nectar.

Histochemical Test	Target Substance	Nectar
PAS reaction	carbohydrates	+
Ruthenium red	acidic mucilage	−
Tannic acid and ferric chloride	mucilage	−
Lugol’s reagent	starch	−
Aniline blue black	proteins	+
Sudan black B	lipids	−
Sudan IV	lipids	−
Nile blue	acidic and neutral lipids	−
Copper acetate and rubeanic acid	fatty acids	−
Ferric chloride	phenolic compounds	+
Ferrous sulphate-formalin	phenolic compounds	+
Dragendorff’s reagent	alkaloids	−
Wagner’s reagent	alkaloids	−

Note. + = present; − = absent.

**Table 2 plants-12-03411-t002:** Identification of compounds, chemical classes, and their respective relative percentual found in the nectar of *Urvillea ulmacea* Kunth.

Compound	Chemical Class	Retention Time (Rt; min)	Relative Percentual (%)
N-acetyl-valine	amino acid	15.73	2.61
8-aminooctanoic acid	amino acid	16.36	11.58
Xylitol	sugar	20.36	19.66
D-fructose	sugar	24.01	21.29
D-glucose	sugar	24.51	17.84
1-Monopalmitin *	lipid	35.82	3.24
Sucrose	sugar	37.35	23.78
	amino acids		14.19
	lipids		3.24
	sugars		82.57

Note: * 1-monopalmitin: glyceryl palmitate.

## Data Availability

All figures and tables of this manuscript have been unpublished and were made specifically for this article.

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
