# Peer review of "Inaugural Description of Extrafloral Nectaries in Sapindaceae: Structure, Diversity and Nectar Composition"

_plants, 2023, doi:10.3390/plants12193411_

Round 1
Reviewer 1 Report
The manuscript submitted for the review presents very interesting research results of the structure and histochemical features of extrafloral nectaries in Urvillea ulmacea (Sapindaceae) and the composition of nectar.
The details of the nectary structure of this species have been illustrated in photographs and described for the first time.
Some minor comments on the title, text, and photograph are presented below:
- I suggest adding "nectar composition" to the title of the manuscript,
as this issue constitutes a substantial part of the study.
- In Results L.70 , there should be : "There are no stomata".
- Fig. 1B - poor photo quality
- In the description of Fig.2, there should be "absence of stomata"
After introduction of the above-mentioned revisions, the paper can be published in Plants.
Minor revision required.
Author Response
The authors thank reviewer's comments. Below we specify the changes, according to the concerns of the reviewer.
Reviewer: I suggest adding "nectar composition" to the title of the manuscript, as this issue constitutes a substantial part of the study.
Authors: We accepted the suggestion and included "nectar composition in the title.
Reviewer: In Results L.70 , there should be : "There are no stomata".
Authors: Changed.
Reviewer: Fig. 1B - poor photo quality
Authors: We have improved the image resolution as best as possible and replaced image 1B in the figure.
Reviewer: In the description of Fig.2, there should be "absence of stomata"
Authors: Changed.
Reviewer 2 Report
Extrafloral nectaries have been previously discovered in plants of Sapindales, however in Sapindaceae species data on anatomical structure of EFNs are very limited. Besides, the questions concerning the chemical composition of a secret and the mechanisms of secretion remain open. Thus, the tasks of presented study were 1) to analyze the structure of extrafloral nectaries, 2) to determine whether there are pores through which secret is released, and 3) to identify the main chemical classes of metabolites in Urvillea ulmacea nectar. The authors were able to discover the unique structure of extrafloral nectaries. Unlike flower nectaries, EFNs lack stomata. The presence of phenolic compounds in nectar is an interesting fact. The presented manuscript is well written, the illustrations are of high quality, and the goals outlined in the work have been achieved. The work may be published without any changes.
Author Response
The authors appreciate the reviewer's comments. No changes were requested.